Structural models for the design of novel antiviral agents against Greek Goat Encephalitis

Papageorgiou Louis 1 2
Loukatou Styliani 1
Koumandou Vassiliki Lila 1
Makałowski Wojciech 3
Megalooikonomou Vasileios 4
Vlachakis Dimitrios 1 dvlachakis@bioacademy.gr
Kossida Sophia skossida@bioacademy.gr 1
1 Computational Biology & Medicine Group, Biomedical Research Foundation, Academy of Athens , Athens , Greece
2 Department of Informatics and Telecommunications, National and Kapodistrian University of Athens , Athens , Greece
3 Institute of Bioinformatics, University of Münster , Münster , Germany
4 Computer Engineering and Informatics Department, University of Patras , Greece
Anisimova Maria
Electronic publication date: 2014 Nov 6
Publication date: 2014
Volume: 2
Electronic Location ID: e664
Received 2014 Jun 18; Accepted 2014 Oct 22
Copyright: © 2014 Papageorgiou et al.
Copyright year: 2014
Copyright holder: Papageorgiou et al.
License: This is an open access article distributed under the terms of the Creative Commons Attribution License, which permits unrestricted use, distribution, reproduction and adaptation in any medium and for any purpose provided that it is properly attributed. For attribution, the original author(s), title, publication source (PeerJ) and either DOI or URL of the article must be cited.
License URL: https://creativecommons.org/licenses/by/4.0/

Keywords: Greek Goat Encephalitis virus, Flaviviridae, RNA-dependent RNA polymerase, Homology modelling, Phylogenetic analysis, Drug design, Helicase, Computational biology

Funding: BIOEXPLORE research project ESF and National Resources Greek national funds COST Action BM1006 Marie Curie Intra European Fellowship This work was partially supported by The BIOEXPLORE research project, which falls under the Operational Program “Education and Lifelong Learning” and is co-financed by the European Social Fund (ESF) and National Resources and ESF and Greek national funds through the Operational Program “Education and Lifelong Learning” of the National Strategic Reference Framework (NSRF)—Research Funding Program: “Thales” Investing in knowledge society through the European Social Fund. The authors are members of the BM1006 COST Action, SeqAhead: Next Generation Sequencing Data Analysis Network. VLK is funded by a Marie Curie Intra European Fellowship within the 7th European Community Framework Programme. The funders had no role in study design, data collection and analysis, decision to publish, or preparation of the manuscript.

==============================
The Greek Goat Encephalitis virus (GGE) belongs to the Flaviviridae family of the genus Flavivirus. The GGE virus constitutes an important pathogen of livestock that infects the goat’s central nervous system. The viral enzymes of GGE, helicase and RNA-dependent RNA polymerase (RdRP), are ideal targets for inhibitor design, since those enzymes are crucial for the virus’ survival, proliferation and transmission. In an effort to understand the molecular structure underlying the functions of those viral enzymes, the three dimensional structures of GGE NS3 helicase and NS5 RdRP have been modelled. The models were constructed in silico using conventional homology modelling techniques and the known 3D crystal structures of solved proteins from closely related species as templates. The established structural models of the GGE NS3 helicase and NS5 RdRP have been evaluated for their viability using a repertoire of in silico tools. The goal of this study is to present the 3D conformations of the GGE viral enzymes as reliable structural models that could provide the platform for the design of novel anti-GGE agents.

Introduction

Greek Goat Encephalitis (GGE) is an endemic virus in Greece which infects the central nervous system of goats and leads to neurological disorders. As an arbovirus (arthropod-borne virus), GGE can be transmitted through ticks. In spite of the association of GGE with Tick-borne encephalitis (TBE) and Louping ill (LI) virus (Pavlidou et al., 2008) which are zoonotic viruses (transmitted from animals to humans), there are no verified zoonosis cases yet regarding the GGE virus (Vlachakis, Tsiliki & Kossida, 2013). However, even though the GGE virus may not infect humans, the results of its epidemics are catastrophic, since national economies are closely related with livestock (Ataide Dias, Mahon & Dore, 2008). GGE is a member of the Flaviviridae virus family (Papageorgiou et al., 2014; Pavlidou et al., 2008), a family with several species causing lethal vertebrate infections, through arthropod vector transmissions (Vlachakis & Kossida, 2013). Numerous important human and animal pathogens are classified in the Flaviviridae virus family. GGE belongs to the genus Flavivirus and specifically the group of Mammalian tick-borne encephalitis viruses (Vlachakis, Argiro & Kossida, 2013). The GGE viral genome is a positive-sense RNA in a linear, single-stranded form, of about 10 kb length.

To date, no anti-GGE virus therapy is available, while many other fatal viruses of the Flaviviridae family also remain untreatable, so there is an urgent need for new antiviral drugs and vaccines (Vlachakis, Tsaniras & Kossida, 2013). Based on recent genomic studies, Flavivirus RNA replication is associated with the non-structural proteins, such as NS3 helicase and NS5 RdRP, encoded by the C-terminal part of the viral polyprotein (Fig. S1) (Nulf & Corey, 2004). Viral RNA-dependent RNA polymerase is encoded by the NS5 domain and catalyzes the replication of RNA. The viral helicase is encoded by the NS3 domain and is the enzyme responsible for the unwinding of the viral genetic material during replication. Therefore, the NS3 viral helicase and NS5 viral RdRP constitute ideal pharmacological targets, as they are key enzymes for the survival, propagation and eventual transmission of the virus (Vlachakis et al., 2012). We propose that structural research and the establishment of a comprehensive in silico platform can aid the de novo design of novel inhibitors (Dalkas et al., 2013).

Methods

Database sequence search

NS3 helicase and NS5 RdRP sequence data from 87 species with fully sequenced genomes were collected from the NIAID Virus Pathogen Database and Analysis Resource (ViPR) (Pickett et al., 2012) (http://www.viprbrc.org), and the NCBI RefSeq database, as described in Vlachakis, Koumandou & Kossida (2013) In total, 7 Pestivirus, 8 Hepacivirus, 3 Pegivirus and 69 Flavivirus sequences were used, as shown in Table S1. In cases where the NS5 annotation was not available in RefSeq, the start and end positions of the NS5 sequence within the whole genome polyprotein was inferred from CLUSTALW alignments with closely related annotated species; only the amino acid sequence corresponding to NS5, determined in this way, was used for the multiple sequence alignments and phylogenetic analysis (Table S1).

Phylogenetic analysis

Alignments of the NS5 protein sequences (available upon request) were created using MUSCLE (Edgar, 2004), and visualized using Jalview (Waterhouse et al., 2009). Only unambiguous homologous regions were retained for phylogenetic analysis; manual masking and trimming was performed in MacClade (Maddison & Maddison, 1989). The NS5 alignment was combined with data about NS3 helicase from our previous work (Vlachakis, Koumandou & Kossida, 2013) to create the tree based on both sequences. For each species, the NS3 helicase and NS5 RdRP sequences were concatenated after alignment, masking and trimming. This concatenated alignment was then tested with ProtTest (Abascal, Zardoya & Posada, 2005) to estimate the appropriate model of sequence evolution. Phylogenetic analysis was performed by Bayesian and Maximum likelihood methods as described in Vlachakis, Koumandou & Kossida (2013) with 100 bootstrap replicates. Trees were visualized using FigTree v1.4 (http://tree.bio.ed.ac.uk/software/figtree/).

Molecular modelling

All calculations and visual constructions were performed on a six-core workstation running Windows 7 and using Molecular Operating Environment (MOE) version 2013.08 software package developed by Chemical Computing Group (Montreal, Canada).

Identification of template structures and sequence alignment

The amino acid sequence of the GGE genome polyprotein was retrieved from the conceptual translation of the virus genome sequence at the NCBI database (http://www.ncbi.nlm.nih.gov/) (genome sequence GenBank: DQ235153.1) (polyprotein sequence GenBank: ABB90677.1). The blastp algorithm (http://blast.ncbi.nlm.nih.gov/Blast.cgi) was used to identify homologous structures by searching in the Protein Data Bank (PDB) (Berman et al., 2002; Berman et al., 2013). The multiple sequence alignment was performed using MOE. Conserved residues in the multiple alignment were visualized with WEBLOGO (http://weblogo.berkeley.edu/logo.cgi).

Homology modelling

The homology modelling of the Greek Goat Encephalitis viral RdRP and helicase were carried out using MOE. The selection of template crystal structures for homology modelling was based on the primary sequence identity and the crystal resolution. The MOE homology model method is separated into four main steps (Vlachakis et al., 2013a). First, a primary fragment geometry specification. Second, the insertion and deletions task. The third step is the loop selection and the side-chain packing and the last step is the final model selection and refinement.

Model optimization

Energy minimisation was done in MOE initially using the Amber99 (Loukatou et al., 2014; Wang, Cieplak & Kollman, 2000) force-field implemented into the same package, up to a root mean square deviation (RMSd) gradient of 0.0001 to remove the geometrical strain. The models were subsequently solvated with simple point charge (SPC) water using the truncated octahedron box extending to 7 Å from the model, and molecular dynamics were performed at 300 K, 1 atm with 2 fsecond step size for a total of ten nanoseconds, using the NVT ensemble in a canonical environment (NVT stands for Number of atoms, Volume and Temperature that remain constant throughout the calculation). The results of the molecular dynamics simulation were collected into a database by MOE for further analysis (Vlachakis et al., 2014).

Model evaluation

The produced models were initially evaluated within the MOE package by a residue packing quality function which depends on the number of buried non-polar side-chain groups and on hydrogen bonding. Moreover, the suite PROCHECK (Laskowski et al., 1996; Papagelopoulos et al., 2014) was employed to further evaluate the quality of the produced models. Finally, Verify3D (Eisenberg, Luthy & Bowie, 1997; Luthy, Bowie & Eisenberg, 1992) was used to evaluate whether the models of NS3 helicase and NS5 RdRP domains are similar to known protein structures of this viral family (Vlachakis, Karozou & Kossida, 2013).

Results

Phylogenetic analysis

Alignment of the Flaviviridae NS5 protein sequences from a variety of Flaviviridae species highlights novel important conserved functional domains (Fig. S2), as described previously for NS3 helicase (Vlachakis, Koumandou & Kossida, 2013). Good conservation is evident throughout the whole length of the sequence, particularly between species that belong to the same genus. The annotated Pestivurs NS5 sequences (∼1215 aa) are significantly longer than the annotated NS5 sequences from Flavivirus (∼903 aa) and Hepacivirus or Pegivirus (∼1039 aa). Pestivirus NS5 shows various internal insertions mostly throughout the N-terminal half of the protein. Regions conserved across all three genera likely indicate important functional domains of the NS5 protein, and the alignment highlights 13 residues which are absolutely conserved between all species. Many of the conserved regions identified here have not been reported previously, and probably deserve further study.

When data for NS3 helicase and NS5 RdRP is combined for phylogenetic reconstruction across all 87 Flaviviridae species (Fig. 1), the results are similar to our previous analysis based on NS3 helicase (Vlachakis, Koumandou & Kossida, 2013). Clear separation of the different genera is seen. Within the Flavivirus, monophylletic groups for the tick-borne and insect-specific species are evident. TABV and the insect-specific Flavivirus appear basal to (closest to the origin of) the whole Flavivirus group. Mosquito-borne species diverge afterwards, followed by the tick-borne species, and species with no known vector (NKV). Most of the groupings in the tree are highly supported by posterior probability and bootstrap values by all three phylogenetic methods.

Figure 1 Phylogenetic reconstruction of Flaviviridae concatenated NS3 helicaseand NS5 RdRP protein sequences.

The tree shown is the best Bayesian topology. Numerical values at the nodes of the tree (x/y/z) indicate statistical supportby MrBayes, PhyML and RAxML (posterior probability, bootstrap and bootstrap, respectively). Values for highly supported nodes have been replaced by symbols, as indicated. Nodes with at least 0.95 posterior probability and 80% bootstrap support were considered robust, and nodes with at least 0.80 posterior probability and 50% bootstrap support are indicated. The position of GGE is indicated by a red arrow. The species for which 3D structure of NS5 RdRP are available are indicated with a blue arrow. The yellow fever virus (YFV), for which the 3D structure of NS3 helicase is available, is indicated with a green arrow. Full details and accession numbers for all protein sequences used are in Table S1. The tree confidently separates the Hepacivurs, Pestivirus, Pegivirusand Flavivirus genera. Within the Flavivirus, TABV and insect-specific species appear basal, whereas Tick-borne species and species with no known vector (NKV) are the most derived.

The tree highlights the phylogenetic distance between species for which structural information regarding NS3 helicase and NS5 RdRP is available. No structure is available from a tick-borne Flavivirus species for either NS3 helicase or NS5 RdRP, so our model proposed here for GGE fills a gap in structural information covering all Flaviviridae lineages. Given that Flaviviridae NS5 sequences for which structures are available come from phylogenetically disparate groups (see arrows in Fig. 1) alignment of these structures allows fine-tuning of the structural model, and highlights conserved domains (Fig. S3).

Structural models of the Greek Goat Encephalitis NS3 helicase and NS5 RdRP

BLASTp against the PDB was used to identify the best available crystal structures as templates for homology modelling. Sequence alignments of the Greek goat encephalitis viral RdRP and viral helicase, to homologs from other Flaviviridae, identified several motifs that are important for biological functionality (Vlachakis, Kontopoulos & Kossida, 2013). A multiple sequence alignment (Fig. S3) was constructed including the Greek Goat Encephalitis NS5 viral RdRP, the crystal structure of Japanese Encephalitis NS5 RdRP with a conserved methyltransferase-polymerase interface (PDB code 4K6M) (Lu & Gong, 2013), the crystal structure of Dengue Virus NS5 RdRP bound to NITD-107 (PDB code 3VWS) (Noble et al., 2013), the crystal structure of Bovine Viral Diarrhea Virus NS5 RdRP (PDB code 1S48) (Choi et al., 2004) and the crystal structure of Hepatitis C NS5 RdRP with short RNA template strand (PDB code 1NB7) (O’Farrell et al., 2003).

The final choice of a template structure was not only based on the percent sequence identity, but also on the structure resolution. The Japanese Encephalitis virus RdRP and Yellow Fever helicase are suitable templates because both viruses belong to the same viral family with Greek Goat Encephalitis (Flaviviridae). The predicted secondary structure for the GGE virus RdRP was found to be similar to the actual structure of the JE virus RdRP; the same was true for the GGE viral helicase and the Yellow Fever viral helicase. The Japanese Encephalitis virus, Dengue virus, Hepatitis C virus and Greek Goat Encephalitis RdRPs all belong to the superfamily of RdRP and share eight common motifs within their domains. The Flaviviridae family helicases belong to the superfamily II of helicases and within their domains share seven common motifs. All RdRP and helicase invariant motifs have been conserved on the Greek Goat Encephalitis models (Lu & Gong, 2013).

The alignment between the RdRP sequence of the Greek Goat Encephalitis and the sequence of the Japanese Encephalitis virus RdRP (PDB 4K6M, resolution 2.6 Å) (Lu & Gong, 2013) template revealed 58% Identity and 73% similarity (Fig. 2). The alignment for the viral helicase between the sequence of the Greek Goat Encephalitis and the sequence of the Yellow Fever virus (PDB 1YKS, resolution 1.8 Å) (Wu et al., 2005) template revealed 50% Identity and 65% similarity (Fig. 3). These results allow for a conventional homology modelling approach to be considered. The two models of NS5 GGE RdRP and NS3 GGE helicase were first structurally superimposed and subsequently compared to their templates using the Swiss Pdb viewer (4.1.0) algorithm (Guex & Peitsch, 1997). The NS5 GGE RdRP model and the NS3 GGE helicase model exhibited an alpha-carbon RMSD of less than 0.78 and less than 1.35 angstroms, respectively. Furthermore, the models were evaluated in terms of their geometry with MOE and the Verify3D algorithm (Eisenberg, Luthy & Bowie, 1997). The compatibility of both the NS5 GGE RdRP model and the NS3 GGE helicase model, with their own amino acid sequence was assessed using the Verify3D algorithm. A structural class was delegated for each residue of every model based on its 3D environment and location (Vlachakis et al., 2013b). Each residue score value was calculated by the algorithm using a collection of reference structures. The scores for the GGE virus RdRP model ranged from + 0.02 to + 0.75 and for the GGE virus helicase ranged from −0.09 to + 0.7. According to these results, the two models can be characterized as having high quality; scores below + 0.1 are indicative of serious problems in the models (Bujnicki, Rychlewski & Fischer, 2002). It was concluded that no poorly scored stretches of amino acids were included neither in structured nor conserved regions of the NS3 helicase and NS5 RdRP models (Table S2 and Fig. S4).

Figure 2 Sequence alignment between the GreekGoat Encephalitis viral NS5 RdRP and the corresponding sequence of the crystal structure of Japanese Encephalitis NS5 RdRP.

All eight major conserved motifs of Flaviviridae have been marked (Motifs A–G).

Figure 3 Sequence alignment between the Greek Goat Encephalitis viral NS3 helicase and the corresponding sequence of the crystal structure of Yellow Fever NS3 helicase.

All seven major conserved motifs of Flaviviridae have been marked (M1, M1a, M2–M6).

Description of the Greek Goat Encephalitis helicase model

As predicted, from the sequence alignment of Greek Goat Encephalitis helicase (Fig. 3) and its secondary structure prediction (data not shown), the constructed model presented the structural features of known Flaviviridae helicases. In the Greek Goat Encephalitis helicase model (Fig. 4) the three distinct domains of helicases were structurally conserved, as well as the various motifs described by Wu et al. (2005). The seven sequence motifs of the GGE helicase occur in domain 1 and 2 (Figs. 3, 4C and 4D) (Gorbalenya & Koonin, 1993). Helicase motif residues are involved in NTP binding and hydrolysis. Moreover, those motifs are involved in coupling the NTPase to RNA duplex unwinding by an unknown mechanism (von Hippel & Delagoutte, 2003). One of the most critical motifs in Flaviviridae helicases is the GxGKT/S motif 1 in domain 1, which in GGE is conserved in the same sequence position (loop) as in kinases. Motif 1 is also known as Walker A motif and its role involves binding of the β-phosphate of adenosine triphosphate (ATP) (Saraste, Sibbald & Wittinghofer, 1990). Based on site directed mutagenesis studies for Motif 1, the mutant protein is inactive. Another crucial conserved motif for the GGE helicase is the DExH motif 2. The DExH motif also known as Walker B motif mediates the binding of the Mg2 +-ATP substrate. The aspartate (Asp 170) was found to bind the Mg2 + and create the best conditions for nucleophilic attack through the optimum orientation of ATP (Ruff et al., 1991). Furthermore, motif 2 is expected to bind a bivalent cation associated with nucleotide γ-phosphate. Finally, another critical motif is the QRxGRxGR motif 5 in domain 2 which is right across the inter-domain cleft from the DExH motif 2. Motif 5 is important to the Flaviviridae helicase function as it is involved in nucleic acid binding (Niedenzu et al., 2001).

Figure 4 NS3 Model of the Greek Goat Encephalitisvirus helicase.

(A) Ribbon representation of the produced Greek Goat Encephalitis virus helicase model. (B) Ribbon representation of the produced Greek Goat Encephalitis virus helicase model (colored Red) next to the corresponding Yellow Fever virus helicase (in green color). (C and D) The conserved motifs of the Yellow Fever virus helicase (in green color) next to the corresponding motifs on the Greek Goat Encephalitis helicase model (colored in Red). The major motifs have been color-coded according to the conventions of Fig. 3, and are shown in CPK format (Usual space filling) along with the rest of the helicase motifs. (E and F) Electrostatic surface potential for the Greek Goat Encephalitis helicase. Represented with red is the area of negative charge. Blue is the area of positive charge and white is the un-charged region.

The Greek Goat Encephalitis helicase model showed a very similar topology to its template, Yellow Fever helicase (Figs. 4B–4D). In general, the Flaviviridae viral helicases contain three domains, which are separated by two channels. Domains 1 and 3 have more interactions with each other than they have with domain 2. During the RNA chain unwinding, domain 2 is expected to undergo crucial movement rather than the other two domains. The channel between domains 1–2 and 3 accommodates the ssRNA during the viral RNA unwinding process. The helicase binds RNA in the specific arginine-rich site contained in the second domain (Luo et al., 2008). Specific contacts between domains 1 and 2 form the helicase motifs. In the NTPase catalytic site these contacts are expected to change, as the domains link together. Motifs 1a and 6 are in direct contact as the two domains are topologically closer. The linker connecting domains 1 and 2 is motif 3. Finally motif 7 forms direct contacts with motifs 1, 2, and 3. The non-structural protein 3 (NS3) helicase has been reported to interact with other viral replication proteins including non-structural protein 5 (NS5) (Filomatori et al., 2006).

Description of the Greek Goat Encephalitis RdRP model

The Greek Goat Encephalitis RdRP secondary structure (data not shown) was predicted from its sequence alignment (Fig. 2). The Greek Goat Encephalitis NS5 model produced of ∼900 amino acids contained two main regions, the N-terminal S-adenosyl-L-methyltransferase dependent methyltransferase (MTase) region and the C-terminal RdRP region (Figs. 5A, 5B and 5D). The MTase region bears a 5′ cap structure and is equally important as RdRP (Geiss et al., 2009). The MTase plays key roles in the capping process and is believed to catalyze the methylation steps and act as a guanylytrasferase to form the linkage (Ray et al., 2006). In the Greek Goat Encephalitis model a linker connects the MTase region and the RdRP region (Figs. 5A, 5B, 5D, 6A1 and 6B). This connection is presented in the Japanese Encephalitis NS5 RdrRP, which is the template model of GGE as well as in other previous studies (Lu & Gong, 2013).

Figure 5 NS5 Model of the Greek GoatEncephalitis virus RdRP.

(A) Ribbon representation of the produced Greek Goat Encephalitis virus RdRP and N-terminal S-adenosyl-L-methyltransferase dependent methyltransferase (MTase) side view model (colored Yellow and green) next to the corresponding Japanese Encephalitis virus RdRP and MTase (in gold and blue color). The linker is shown in red. (B) Ribbon representation of the produced Greek Goat Encephalitis virus RdRP (colored green) and MTase (colored yellow) side view complex model. The linker is shown in red. (C and D) Electrostatic surface potential side view for the Greek Goat Encephalitis RdRP and MTase complex.

Figure 6 Global views of Greek Goat Encephalitis NS5 RdRP shown in different orientations.

(A1) Ribbon representation of the produced Greek Goat Encephalitisvirus RdRP model, looking into the RdRP front channel (putative dsRNAchannel). The three main domains of RdRP are shown: thumb (colored violet), palm (gray) and finger (light blue), with the MTase region at the back (yellow). The linker is shown in red. (A2) Ribbon representation of the produced Greek Goat Encephalitis virus RdRP model with more details about the finger domain subdomains, looking into the RdRP front channel (putative dsRNA channel); (B) Sideview of Ribbon representation of the produced Greek Goat Encephalitis virus RdRP model with more details about the finger domain subdomains. The linker between MTase and RdRP is shown in red.

Flaviviridae non structural protein 5 (NS5) comprises eight conserved motifs (Lu & Gong, 2013) (Fig. 7). The three domains of Greek Goat Encephalitis RdRP were found to be structurally conserved, as well as the various motifs and regions of Flaviviridae NS5 (Fig. 2). As with other viral Flaviviridae RdRP, Greek Goat Encephalitis RdRP is separated into the N-terminal extension, the main polymerase and the priming loop. The main polymerase adopts a shape analogous to a cupped right hand and contains the finger and thumb domains, which rise on the sides of the palm domain (Fig. 6A1). The most conserved part of viral RdRP is the palm domain, which contains the motifs A, B, C and D (Lu & Gong, 2013). In motifs A and B there are two catalytic aspartic acid residues (D536 and D668) that are conserved in all single-subunit possessive polymerases (Gong & Peersen, 2010). The finger domain contains the motifs F and G. Despite the fact that these motifs have not been resolved, these RdRP structures exhibit high similarity to the NS5B that are represented in the other two genera of the Flaviviridae family (Gong & Peersen, 2010). The thumb domain of Greek Goat Encephalitis viral RdRP is relatively divergent and contains only Motif E. The size of the thumb domain is larger than in other viral Flaviviridae RdRPs because it carries additional elements (C-terminal extension) that facilitate de novo initiation.

Figure 7 Global views of Greek Goat Encephalitis NS5 RdRP shown in different orientations.

The major motifs have been color-coded according to the conventions of Fig. 2, and are shown in CPK format (Usual space filling).

The Greek Goat encephalitis RdRP finger domain is separated into individual subdomains according to Japanese Encephalitis (Lu & Gong, 2013) and Powassan Encephalitis RdRP (Thompson & Peersen, 2004). In the finger domain four different subdomains have been marked, the index finger, the middle finger, the ring finger and the pinky finger (Figs. 6A2 and 6B). The index finger comprises a nuclear localization signal that coincides with the recommended binding site of the non-structural protein 3 (NS3) (Rawlinson et al., 2009). Moreover the thumb domain interacts with the tip of the index finger and forms a unique encircled active site. The second and the third strands of the fingers domain are in the middle finger area. No specific function has been related with the middle finger area yet. The forth and the fifth strands of the fingers domain are described as ring finger and this area includes the NTP binding of motif F. Finally the pinky finger contains motif G, which is relatively bulky and forms one side of the dsRNA channel (Gong & Peersen, 2010).

Conclusions

The 3D models of the Greek Goat Encephalitis viral enzymes were designed using homology modelling. The X-ray crystal structure of the viral Yellow Fever helicase was employed as a template for the Greek Goat Encephalitis viral helicase and the Japanese Encephalitis RNA dependent RNA polymerase as a template for the Greek Goat Encephalitis RNA dependent RNA polymerase. The models were evaluated and display high conservation of the functional domains previously characterized in other Flaviviridae species. We therefore nominate our Greek Goat Encephalitis enzymes models to be suitable for advanced, in silico de-novo drug design experiments. In the future this drug discovery process may lead to the development of potential inhibitor molecules.

Supplemental Information

Figure S1 Schematic diagram of the structural and non-structural proteins within the GGE genome polyprotein.

The different complexes that are thought to arise in different cellular compartments during and following polyprotein processing are shown underneath.

Click here for additional data file.

Figure S2 Alignment of Flaviviridae NS5 protein sequences.

The alignment was generated using Muscle, and visualized with Jalview. Amino acids are colored blue based on percent identity in the alignment, the consensus weblogo is shown at the bottom.

Click here for additional data file.

Figure S3 Multi-alignment of Flaviviridae NS5 protein sequences

Click here for additional data file.

Figure S4 GGE NS3 Helicase and NS5 RdRP models geometry assessment

Click here for additional data file.

Table S1 Details and accession numbers of all sequences used for the alignments and phylogenetic analysis

The table also indicates species names abbreviations used in Figs. 8 and 7.

Click here for additional data file.

Table S2 Model description and data validation

Click here for additional data file.

Supplemental Information 7 Greek goat encephalitis virus NS3 Helicase model

.pdb file

Click here for additional data file.

Supplemental Information 8 Greek goat encephalitis virus NS5 RdRP model

.pdb file

Click here for additional data file.

We are indebted to the CamGrid computational resource on which the phylogenetic analyses were performed. LP is grateful to Prof. Maroulis Dimitrios for his continuous support throughout his PhD research.

Additional Information and Declarations

Competing Interests

Author Contributions

The authors declare there are no competing interests.

Louis Papageorgiou performed the experiments, analyzed the data, wrote the paper, prepared figures and/or tables.

Styliani Loukatou performed the experiments, analyzed the data, wrote the paper.

Vassiliki Lila Koumandou performed the experiments, analyzed the data, wrote the paper, prepared figures and/or tables, reviewed drafts of the paper.

Wojciech Makałowski and Sophia Kossida conceived and designed the experiments, reviewed drafts of the paper.

Vasileios Megalooikonomou conceived and designed the experiments, analyzed the data, reviewed drafts of the paper.

Dimitrios Vlachakis conceived and designed the experiments, performed the experiments, analyzed the data, wrote the paper, prepared figures and/or tables, reviewed drafts of the paper.

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
