# Peer review of "Structural models for the design of novel antiviral agents against Greek Goat Encephalitis"

_PeerJ, doi:10.7717/peerj.664_

## Round 0.1 · original submission · Minor Revisions

Please address the comments of the reviewers.

Reviewer 1 ·

Basic reporting

No comments

Experimental design

No comments

Validity of the findings

Minor issue, please provide more details about the results on lines 180-183 in the manuscript. Specifically, what percentage of the residues in each protein model have scores less than +0.1 (and especially less than 0), and where are they located in the proteins? Are these poorly scored positions located within coils exclusively, or do they appear in alpha helices and beta strands as well? Are they located within any particular motifs? Answers to these questions will help provide a fuller picture of the models.

Reviewer 2 ·

Basic reporting

The manuscript is generally well written and data is clearly presented. As a suggestion, I think the description of the models could be more concise and the validation data should be presented in a table or a figure.

Experimental design

The methodology reported is relatively straight forward and the results seem to be accurate. My only concern regards the selection of the templates. Although I do not dispute the validity of using the YFV proteins, it is incorrect to state that only one NS3 from flaviviruses has been crystallised. In fact, beside YFV, there are several DENV helicases and full NS3 protease/helicase structures and a JEV helices structure available (and possibly more). On a similar note, the authors should always refer to the NS3 helicase in the text and not simply NS3, as this protein has two functional domains.
Furthermore, in the case of the NS5 templates, BVDV is not a flavivirus, but a pestivirus.

Validity of the findings

The models generated are interesting and indeed I would the authors to attach the coordinates of the structures as supplemental information

---

## Round 0.2 · accepted · Accept

Thank you for carefully addressing the reviewers comments.